# Considerations for Evaluating the Introduction of New Cancer Screening Technology: Use of Interval Cancers to Assess Potential Benefits and Harms

**DOI:** 10.3390/ijerph192214647

**Published:** 2022-11-08

**Authors:** Rachel Farber, Nehmat Houssami, Isabelle Barnes, Kevin McGeechan, Alexandra Barratt, Katy J. L. Bell

**Affiliations:** 1Wiser Healthcare, Sydney School of Public Health, Faculty of Medicine and Health, The University of Sydney, Sydney 2006, Australia; 2The Daffodil Centre, The University of Sydney, A Joint Venture with Cancer Council NSW, Sydney 2006, Australia; 3Sydney School of Public Health, Faculty of Medicine and Health, The University of Sydney, Sydney 2006, Australia; 4Centre for Women’s Health Research, College of Health, Medicine and Wellbeing, The University of Newcastle, Callaghan 2308, Australia; 5Australian Longitudinal Study on Women’s Health, The University of Newcastle, Callaghan 2308, Australia

**Keywords:** cancer, cohort, ductal carcinoma in situ, mammography, recall, breast mammography, digital, health outcomes, film screen, interval cancer

## Abstract

This framework focuses on the importance of the consideration of the downstream intermediate and long-term health outcomes when a change to a screening program is introduced. The authors present a methodology for utilising the relationship between screen-detected and interval cancer rates to infer the benefits and harms associated with a change to the program. A review of the previous use of these measures in the literature is presented. The framework presents other aspects to consider when utilizing this methodology, and builds upon an existing framework that helps researchers, clinicians, and policy makers to consider the impacts of changes to screening programs on health outcomes. It is hoped that this research will inform future evaluative studies to assess the benefits and harms of changes to screening programs.

## 1. Introduction

Cancer screening programs aim to reduce morbidity and mortality through the early detection and treatment of cancers that would have otherwise have gone on to present at a later stage [1]. When a change is introduced to a screening program, such as a change in technology, it is important to evaluate whether this change results in an overall net benefit. The potential benefits from the change should not look only at the initial impact on improved sensitivity, but also evaluate whether this improved sensitivity addresses clinically important cancers where early detection is, providing outcomes which are likely to translate into longer-term benefits [2]. While it is important to estimate the increases in screen-detected cancers, we need to know the extent to which the additional cancers detected are those that would have otherwise progressed to present later, representing potential beneficial early detection—versus indolent lesions that would not have progressed at all, or would only have grown very slowly—representing potential harm from over-detection.

## 2. Role of Interval Cancers

The prompt assessment of the effects of new technology or screening protocols requires methods using short-term outcomes [3]. Once a screening program is underway, interval cancers can be used as a proxy measure for cancers not currently being detected by the existing screening program [4,5]. Interval cancers are cancers diagnosed after a negative screening result and before a subsequent scheduled screening. Interval cancers may not be detected during screening either because they: did not yet exist, were not detectable with the modality used or were missed by the person reading the screens. Most interval cancers are symptomatic and are therefore by definition clinically important. They have been found to have similar prognostic features to other cancers clinically diagnosed outside of screening [6]. Interval cancers tend to be more biologically aggressive tumours with a faster growth rate and more unfavourable tumour characteristics than screen-detected cancers [7,8,9]. Therefore, interval cancers represent the cancers where early detection would be beneficial.

For a change to a screening program to increase its effectiveness, it should result in an increase in the diagnosis of clinically important early stage cancers, a decrease in interval cancers diagnosed between scheduled screenings, and ultimately a decline in the incidence of cancers diagnosed at late stage. However, if such changes are not seen, any additional cancers diagnosed via a change to the program may represent overdiagnosed cancers that would not have otherwise presented, or more slowly growing cancers that would have been detected in the old screening program without detriment [3,10]. The emphasis in screening should not be on finding more cancers, but on finding more cancers that are clinically important [11]. The tension between maximising benefit through the detection of clinically important cancers and minimising harm through the detection of clinically unimportant lesions, means that screening programs ought to undergo regular re-evaluation, particularly when changes are introduced [12,13]. It is important to ensure that any changes in the implementation of screening optimize the balance between benefit and harm, including minimising the risk of overdiagnosis [14].

## 3. Relationship between Screen-Detected and Interval Cancers

A reduction in interval cancers is sometimes used as a surrogate measure of a change in screening effectiveness [5]. Previously, screen-detected cancer rates and interval cancer rates have been used to measure the benefit from a new intervention. However, it is also important to consider the reverse implication of an increase in detection rate without a meaningful decrease in interval cancer rates. Our framework takes the relationship between interval cancers and screen-detected cancers a step further, to consider how this relationship might be used as both an indicator of benefit, and of harm, from changes made to the program.

Figure 1 shows the possible scenarios concomitant upon a change in technology. If a change to a new technology increases sensitivity for clinically important cancers we expect to see:(1)an increase in screen-detected cancers with tumour markers indicating aggressive disease; and(2)a decrease in the interval cancer rate between screenings, indicating that at least some of the additional detection is occurring in clinically important and rapidly progressing cancers.

However, if the change only increases the detection of clinically unimportant cancers, we expect to see:(3)an increase in the detection of screen-detected cancers with tumour markers indicating low risk disease; and(4)no or minimal changes in the interval cancer rate between screenings, indicating that the additional detection is of overdiagnosed or slow progressing cancers.

The relationship between screen-detected and interval cancers can be further identified by plotting the risk difference in screen-detected and interval cancer rates before and after the change. Rates are typically presented per 1000 screens.

In Figure 2 each quadrant represents a different signature:(1)The top left quadrant shows a decrease in screen detection and an increase in interval rates. This observation indicates that there is decreased rate of detection, including clinically important cancers, and therefore suggests an increase in underdiagnosis(2)The bottom left quadrant shows a decrease in screen detection and a decrease in interval rates. This result indicates that there is decreased detection overall, but an increased detection of clinically important cancers, and therefore suggests a decrease in overdiagnosis. Theoretically this could occur with risk-stratified screening [15].(3)The top right quadrant shows an increase in screen detection and no change, or possibly even an increase, in interval rates. Assuming no change in the background rates of cancer, this observation would indicate that the increased detection represents an increase in overdiagnosis.(4)The bottom right quadrant shows an increase in screen detection and a decreased interval rate. This finding indicates that the increased detection does include clinically important and progressing cancers, and therefore suggests a decrease in underdiagnosis. The second and fourth quadrants show the improved effectiveness sought with a change to the screening program.

It is important to look not only at the relationship between screen-detected and interval cancer rates, but also at the tumour characteristics of the cancers detected. Differences between technologies in both screen-detected cancers and interval cancers for tumour characteristics such as size, histological type, grade, node status, and stage, allow for further assessment of the prognostic changes in the cancers detected [16].

## 4. Previous Uses of Interval Cancers as an Indicator of Benefits and Harms

We conducted a targeted review (R.F. & I.B.) of the literature to examine the way in which interval cancer rates have been used to assess the harms and benefits of changes to screening programs, to identify if similar applied methodologies used interval cancers, as described above. (For search strategy see Figure 3, and for selection of studies see Figure 4) We identified studies that reported on an asymptomatic population at normal risk of the relevant cancer, which reported on either a change to a screening program/technology or to a direct comparison between two or more screening programs/technologies, and assessed both interval cancer rates and screen-detected cancer rates. A summary of the scoping review is presented in Table 1.

Most of the included studies in this review reported on changes to breast screening programs and technologies [10,17,19,20,21,22,23,24,25,26,27,28,29,31,32,34,35,37,38,40,41,42,43]. Other cancer screening programs examined included gastric cancer [36], colorectal cancer [18,33], and prostate cancer [39]. While most studies compared two types of screening technology approaches (e.g., film versus digital mammography), some studies utilised screen-detected and interval cancer rates to determine optimal screening interval time-frames (e.g., annual versus two-yearly screening intervals) [39,41].

Studies that reflected on the relationship between screen-detected and interval cancers have often used this relationship as an indicator of benefit, but few have considered it as an indicator of harm [4,5,44]. Of the included studies, 11 consider overdiagnosis, however all but four conclude the results reflect it is not of concern [17,21,22,39]. Some studies utilised the relationship between screen-detected and interval cancers as a measure of effectiveness. For example, McDonald et al. found an increase in screen-detected cancer rates and a subsequent decrease in interval cancer rates for digital breast tomosynthesis compared to digital mammography, and noted this as a surrogate outcome for screening benefit [28]. Other studies found the lack of change in screen-detected or interval cancer rates as an indicator of no additional benefit. Lehman et al. utilised the comparison between screen-detected cancer and interval cancer rates to assess the addition of computer-aided detection (CAD) in digital screening mammography. Considering the increased financial burden on women who opt for mammographic screening with CAD, the authors were able to conclude that CAD offers no increase in clinical benefit [31].

The limitations of sole reliance on interval cancers were noted with the importance of complementing the comparison of interval and screen-detected cancers with tumour characteristics, to ensure increases in detection rate are reflecting clinically important cancers [7,37]. In most studies comparing interval cancers with screen-detected cancers, there are consistent findings indicating that interval cancers have worse prognostic features, such as larger tumour size, higher frequency of node metastases, higher histologic grade, and more advanced disease than screen-detected cancers [6].

One example is a study that used interval rates to consider both benefits and harms, as does our framework, comparing tomosynthesis to digital mammography [21]. This study found that a substantial increase in screen-detected cancers did not lead to a commensurate reduction in interval cancers. The authors noted this finding raises the possibility of overdiagnosis, but commented that their study was not sufficiently powered to measure an effect on interval cancer. Nevertheless, they concluded it seemed reasonable to suspect that some of the increased detection from tomosynthesis is contributing to overdiagnosis in population breast cancer screening [21]. Another study proposed that the lack of translation from detection rate to interval cancer rate may be due to a delay in lead time and/or to overdiagnosis [17].

## 5. Other Considerations

It is important to consider underlying cancer rates in the population, as they can affect the screen-detected and interval cancer rates. Comparisons of interval cancer should be limited to monitoring within screening services or programs, because comparisons between screening programs and countries is limited by heterogeneity [6]. Changes in screening programs often involve a before/after comparison within a single population, rather than a comparison with a control group, and are thus at higher risk of bias. When comparing interval cancer rates to screen-detected cancer rates to measure a change in a screening program, it is important to consider other factors that may have affected these rates, beyond the change of interest. One way to adjust for these temporal changes that may be confounding the relationship is to not only compare overall rates using methodology such as interrupted time series to limit the attribution to the change of interest. Using an interrupted time series limits the selection bias and confounding due to between-group differences [45,46,47]. However, there may still be challenges to the comparison, even after accounting for background changes over time. One challenge is how best to account for cancers in people who do not comply with rescreening recommendations. Some people who attended at least one screening have a cancer diagnosed beyond the recommended rescreening time. These cancers, which are neither screen detected nor interval, are not included in the Framework’s comparison. Care is needed to ensure that omission of these cancers from analysis does not cause a biased estimate of the difference in interval rates.

## 6. Learning Curves

The changes in screening programs have been associated with an initial increase in false positive rates [48,49]. This change is probably due to adjustment periods, with aspects like new technologies, additional views, or changes in the number of readers [50]. While the increase may be short lived, the impact on those people who were sent down the clinical pathway of being recalled for further investigation has been shown to be associated with significantly adverse psychological effects in both the short and long term [48,51]. Depending on how drastic the change is, even if an increase in false positives is relatively brief, recall rates may take time to stabilise, and this factor must be considered in relation to the frequency at which a new technology is introduced.

## 7. Other Benefits and Harms

In addition to the impact of changes in screening programs on health outcomes, it is important to consider the workflow, economic, and environmental cost-effectiveness of such changes. An example of this cost-effect analysis is when a change in technology has significantly shortened the acquisition time but has significantly increased the interpretation time [52]. The initial and ongoing cost impact of a change may be in opposite directions. When this is the case, it is important to determine the duration; the change needs to be sustained to outweigh the upfront costs [53]. These considerations should include both the initial impact of the change, for example the environmental cost of a new screening technology, and the ongoing cost reduction in the digitalisation of screens [54,55].

## 8. Future Changes

The design and implementation of screening programs around the world will continue to change. It is human nature to think that new technology is better, but the translation to improved health outcomes needs to be evaluated [56]. The measurement of interval cancer rates needs to be prioritised in studies evaluating changes in technology, as they are often a recognised evidence gap [21]. The measurement of interval cancers can be used in screening evaluation, linked to data from cancer registries or obtained through academic trials [57]. Emerging areas in screening, such as artificial intelligence, could also benefit from the application of this framework [58]. Often changes are first introduced in symptomatic clinics; while this information is important in evaluating a change, the probability of disease in a symptomatic person may differ substantially from that of an asymptomatic person [59]. Interval cancers should be considered not only in planned changes to cancer screening, but also in unplanned changes, such as when the COVID-19 pandemic forced a move to telehealth models of healthcare [60]. As new evidence emerges, it is important that decision aids are updated with information relevant to the condition of that screening program [61].

## 9. Conclusions

This framework puts forward a methodology of utlising intermediate outcomes of screen-detected and interval cancers rates to contextualise and weigh the indicators of benefits and harms of changes to a screening program. This approach is important for the consideration of changes in cancer screening practice and policy accounting for both short- and intermediate-term health outcomes.

## Figures and Tables

**Figure 1 ijerph-19-14647-f001:**
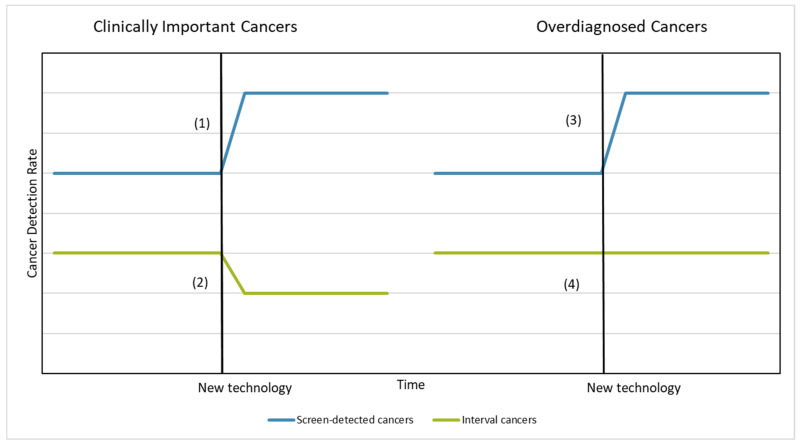
Potential scenarios following the introduction of a new technology to a screening program.

**Figure 2 ijerph-19-14647-f002:**
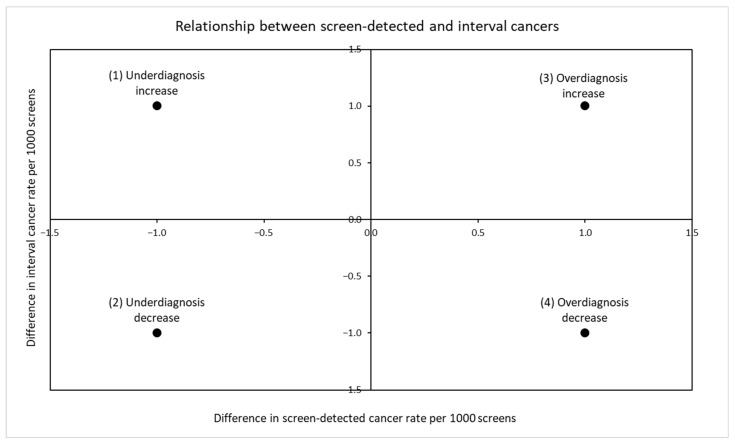
Indications from the relationship between screen-detected and interval cancer rates subsequent to a change in a screening program.

**Figure 3 ijerph-19-14647-f003:**
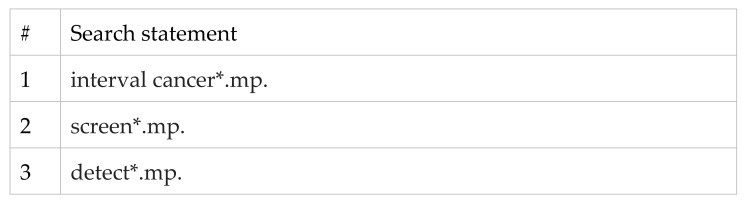
Medline Search Strategy. asterisk (*) represents a truncation search.

**Figure 4 ijerph-19-14647-f004:**
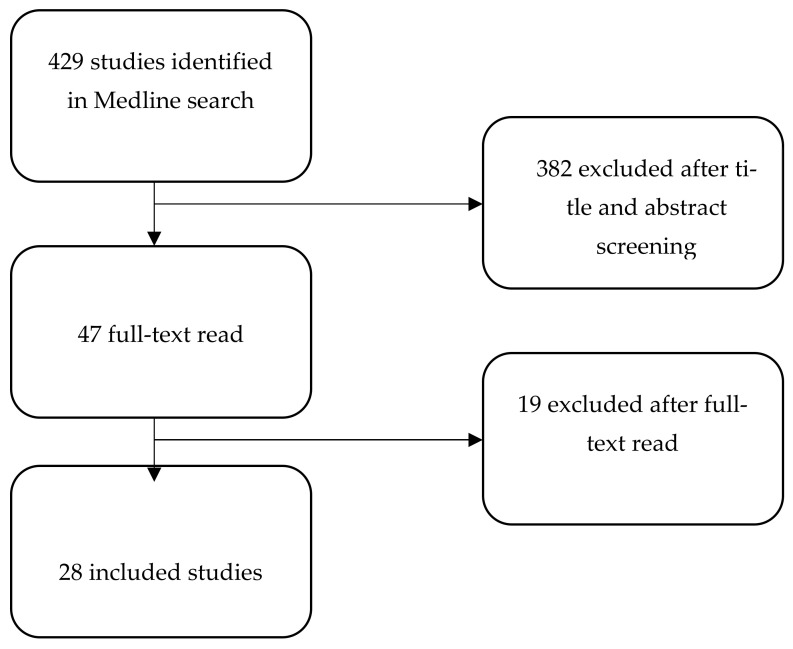
Selection of included studies.

**Table 1 ijerph-19-14647-t001:** Summary of selected studies from scoping review.

Reference	Study Design	Country	Study Time Points	Change/Comparison	Screen-Detected Cancer Rate	Interval Cancer Rate	Harms and/or Benefits
Armaroli (2022) [17]	Randomised control trial	Italy	2014–2020	Digital mammography (DM) versus digital breast tomosynthesis (DBT)	Significantly higher with DBT	Did not differ significantly between the two arms	Compatible either with greater lead time gain with DBT screening in comparison with DM screening and/or with an excess of detection of indolent cancers (overdiagnosis) in the DBT arm.
Bretagne (2021) [18]	Cohort	France	2010–2017	Faecal immunochemical testing (FIT) versus guaiac faecal occult blood testing (gFOBT).	Higher screen-detected ancer rate with FIT	Lower interval cancer rate with FIT	Dramatic decrease in the cumulative incidence rates of interval cancers after switching from gFOBT to FIT. These data provide an additional incentive for countries still using gFOBT to switch to FIT.
Hofvind (2021) [19]	Randomised Control Trial	Norway	2016–2020	DBT versus DM	No significant difference in screen-detected breast cancer among women screened with DBT versus DM	No difference in interval cancer rate with DBT	DBT benefits were not at the cost of worse subsequent round screening outcomes.
Johnson (2021) [20]	Cohort	Sweden	2010–2015	one-view DBT and two-view DM versus two-view DM	Sensitivity was higher for DBT than for DM	one-view DBT and two-view DM was lower than DM	Effect of DBT on interval cancer rate in population screening and that it could translate into additional screening benefit
Bernardi (2020) [21]	Cohort	Italy	2013–2016	DBT versus DM	DBT significantly improved early detection measures	Did not significantly reduce ICR (relative to DM screening),	Not well powered for interval cancers but suggesting that it could add benefit as well as adding over-detection in population BC screening.
Hovda (2019) [22]	Cohort	Norway	2008–2016	DBT versus DM	Higher for DBT	No difference in rates	Findings show that the excess cancers diagnosed with DBT seem to be less aggressive tumors, indicating potential overdiagnosis; cannot conclude about the clinical implications as the study was not designed to investigate overdiagnosis; further studies are needed; interval breast cancer rate examination is essential when evaluating screening effectiveness
Bahl (2018) [23]	Cohort	US	2009–2011 (DM); 2013–2015 (DBT)	DM versus DBT	No difference in rates; higher proportion of screen-detected cancers were invasive rather than in situ when compared with DM	No difference in rates; MRI-detected interval cancers were more likely to be minimal cancers, compared to symptomatic interval cancers	Complete integration of DBT saw higher rates of screen-detected cancers which were invasive and lower rates of screen-detected in situ cancers; although no change in interval cancer rate between screening technologies, a higher proportion of screen-detected cancers are invasive with DBT.
Houssami (2018) [24]	Cohort	Italy	2011–2012	Tomosynthesis 3D-screening versus standard 2D-mammography alone	Screening sensitivity higher for integrated 2D/3D mammography	Lower for tomosynthesis	Much larger screening studies or pooled analyses are necessary in order to examine interval cancer rates arising; small number of interval cancers precludes analytic evaluation and the study was not planned for interval cancer rate comparison, thus results must be interpreted with caution;
Sankatsing (2018) [25]	Cohort	Netherlands	2004–2011	SFM to DM	Increased screen-detected cancer rate for DM	No difference in rates	There is warranted concern that increase in detection of cancers with DM is indicative of overdiagnosis, however, it was demonstrated that invasive cancers were more likely to be detected
Skaane (2018) [26]	Randomised controlled trial	Norway	2010–2012	DM plus DBT versus DM alone	Higher screen-detected cancer rate for DM+DBT	No difference in rates	Beneficial to distinguish between missed and true interval cancers, which this study did not do; a study with longer follow-up periods and more screening rounds would be required to estimate harms in the form of overdiagnosis; it is important to remember that overdiagnosis does not diminish the benefits of mammography in reducing breast cancer mortality
de Munck (2016) [27]	Cohort	The Netherlands	2004–2010	Screen-film mammography (SFM) versus DM	No difference in rates	No difference in rates	DM can safely be used for screening; DM facilitates easier image transfer and may increase efficiency in hospitals; no indicator of overdiagnosis with shift to DM
McDonald (2016) [28]	Cohort	US	2010–2014	DM versus DBT	Increased screen-detected cancer rate for DBT	Decreased interval cancer rate for DBT	Digital breast tomosynthesis was associated with increased cancer detection and decrease interval cancers; it is unlikely that insignificant cancers were detected at an earlier stage (potential overdiagnosis) as the invasive cancer detection rate remained stable; the primary harm of screening was reduced with decreased false-positive examinations
Prummel (2016) [29]	Cohort	Toronto	2008–2009	Digital computed radiography (CR) versus direct radiography (DR), versus SFM for breast cancer	Lowest screen-detected cancer rate for CR; screen-detected cancer rates for DR and SFM were similar	Highest interval cancer rate for CR; similar interval cancer rates for DR and SFM	Significantly lower cancer detection for CR mammography suggests that CR screening is missing a large number of breast cancers, in comparison with DR and DFM; programs need to monitor the performance of CR separately and assess its continued use over time
Sverzellati (2016) [30]	Randomised controlled trial	Italy	2000–2011	Comparison of two different strategies of lung cancer screening by low-dose computed tomography (LDCT), namely, annual (LDCT1) or biennial (LDCT2) screen.	No difference in rates	No difference in rates	Biennial screening may be at least as efficient as annual screening in terms of screen detection rates and other outcomes (i.e., sensitivity, specificity); reducing the number of LDCT screens is less harmful to patients and more cost-effective
Klompenhouwer (2015) [10]	Cohort	The Netherlands	2009–2011	Blinded double reading versus non-blinded double reading in a biennial mammography screening population	Higher for blinded double reading	Lower for blinded double reading	Tumour characteristics were examined and were comparable between screening strategies; data suggest that potential overdiagnosis does not differ between blinded and non-blinded double reading; cost-effectiveness harms and benefits still required to evaluate screening holistically; cancer mortality is the best measure of screening effectiveness, but requires long-term follow up
Lehman (2015) [31]	Cohort	US	2003–2009	Digital screening mammography with and without computer-aided detection (CAD)	No difference in rates	No difference in rates	No increased benefit of CAD to women; suggestion that insurers pay more for CAD for no clinically beneficial reason
Sala (2015) [32]	Cohort	Spain	1995–2012	SFM to DM	Increased screen-detected cancer rate for DM	No difference in rates	Increased detection rate with the introduction of DM is in part due to earlier diagnosis, however, examination of cancers showed that invasive cancers were more commonly detected DM; the use of digital technology should not be assumed a threat which increases overdiagnosis in screening
Chiang (2014) [33]	Cohort	Taiwan	2004–2009	Two different FITs for colorectal cancer screening: OC-Sensor versus HM-Jack	Lower screen-detected cancer rate for HM-Jack	Higher interval cancer rate for HM-Jack	Efforts to improve the effectiveness of FITs can be measured using the occurrence of interval cancers; other indicators of performance are needed which assess both short-term and long-term outcomes
Dibden (2014) [34]	Cohort	UK	2003–2005	One-view versus two-view mammography	Higher for two-view mammography	Lower for two-view mammograpy	A reduction in interval cancer rates was associated with two-view mammography; the increase in screen-detected cancer associated with two-view mammography is not likely to be due to overdiagnosis
Hofvind (2014) [35]	Cohort	Norway	1996–2010	SFM versus DM	No difference in rates	No difference in rates	After initial transitional phase, DM saw lower recall and biopsy rates and thus was associated with less harm; relatively small number of cases means studies generally do not including interval cancer rates; larger population-based studies with sufficient follow-up periods are necessary for screening evaluations;
Choi (2012) [36]	Cross-sectional	Korea	2002–2005	Upper-gastrointestinal series (UGIS) versus endoscopy screening for gastric cancer	Higher screen-detected cancer rate for endoscopy	No difference in interval cancer rate	Endoscopy performed better than UGIS with indicators that it may have a larger impact on gastric cancer mortality, however this was not certain; could not distinguish between false-negative and true interval cancers
Hoff (2012) [37]	Retrospective cohort	Norway	2002–2008	SFM versus DM	No difference in rates	No difference in rates	The transition to DM did not reduce the challenge of missed cancers; characteristics differed between cancers missed at SFM and DM; it is necessary to perform analyses which report on the mammographic features in all missed cancers; study differentiated between true and missed cancers
Seigneurin (2009) [38]	Cohort	France	1994–2004	Two-view versus single-view screening mammography	Higher screen-detected cancer rate for two-view mammography	Lower interval cancer rate for two-view mammography	Findings reveal benefits of performing two-view mammography; decrease in interval cancer rate paralleled by increase in the cancer detection rate;
Roobol (2007) [39]	Randomised controlled trials	The Netherlands and Sweden	1993–1995 (Rotterdam study); 1999–2005 (Gothenburg study)	2-year versus 4-year screening intervals for prostate cancer	Higher for 2-year screening interval	No difference in rates	There is potential overdiagnosis for 2-year prostate cancer screening, due to seeing no difference in the interval cancer rate between interval lengths; the appropriateness of length of screening intervals in cancer screening programs can be assessed using interval cancer rates
Skaane (2007) [40]	Randomised controlled trial	Norway	2000–2001	SFM versus DM	Higher screen-detected cancer rate for DM	Lower interval cancer rate for DM	DM has several potential benefits in mammographic screening, not limited to clinical outcomes; reducing interval cancer rates is crucial in breast cancer screening as it represents the benefits of early detection; further studies are needed to make conclusions about harms and benefits
Wai (2005) [41]	Cohort	British Columbia	Prior to 1997 and after 1997 (exact years not reported)	1-year and 2-year screening intervals for women undergoing screening mammography	Higher screen-detected cancer rate for annual mammography screening	No difference in rates	Other measures needed to assess improvement (i.e., prognostic models showing survival rates)
Hunt (1999) [42]	Retrospective cohort	US	1985–1997	Annual versus biennial screening mammography	Lower for annual screening	Lower for annual screening	Low numbers meant that trends did not reach statistical significance; further research needed to examine harms and benefits
Warren (1997) [43]	Cohort	UK	1989–1991	A comparison of the effectiveness of 28 kV (grid) versus 25 kV (no grid) mammographic techniques for breast screening	Higher screen-detected cancer rate for 28 kV grid	Higher interval cancer rate for 28 kV grid	Non-significant difference in cancer detection rate was offset by an increase in interval cancer rate

## Data Availability

Not applicable.

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
