# Peer review of "Considerations for Evaluating the Introduction of New Cancer Screening Technology: Use of Interval Cancers to Assess Potential Benefits and Harms"

_ijerph, 2022, doi:10.3390/ijerph192214647_

Round 1

Reviewer 1 Report

This manuscript discussed the current methodology for utilizing the relationship between screen-detected and interval cancer rates to evaluate the introduction of new cancer screening technologies and the new framework was proposed. It was stated that it is important to look not only at the relationship between screen-detected and interval cancer rates, but also at the characteristics of the cancers detected.

Overall, this is a comprehensive review with novel and significant findings. There are only few concerns to be clarified before publication.

First, there are a few technical issues, such as the use of references (reference numbers should be before the full stop) or the use of the font size (in the last part ‘References’).

In the part ‘Role of interval cancers’, there are some repetitions, for example that interval cancers are usually symptomatic and more likely to be progressive. Please, modify that part in order not to have unnecessary repetitions.

In Figure 2 there is no legend, and in Figure 1 it is not clear what the legend is (if the legend includes the explanation of all four scenarios or not). Besides, it is not clear in Figure 1 why the increase and the decrease of cancer detection rates are before new technology implementation, and not after.

At page 4, there is a part of the text ‘We conducted a targeted review (RF & IB) of the literature…’. In my opinion, the contribution of the authors in this part is redundant (there is ‘author statement’ at the end).

Table 1 is comprehensive but it is not easy to follow due to a large amount of information. I suggest to add only reference number instead of the whole reference, it would make the table easier to follow.

At page 16, please define CAD here and specify the number of studies that considered the relationship between screen-detected and interval cancers as an indicator of harm (instead of ‘few’).

At the end, in my opinion, there should be the ‘Conclusion’ section, to be more specific about the proposed new framework to evaluate the introduction of new cancer screening technologies, which is the main aim of the manuscript.

Reviewer 2 Report

Author’s suggestions

The present paper by Rachel Farber et al focuses on the importance of the consideration of the downstream inter-mediate and long-term health outcomes in respect to a screening program. The authors present a methodology for utilizing the relationship between screen-detected and interval cancer rates to infer the benefits and harms associated with a change. The manuscript is written very well and could be accepted after the incorporation of following minor correction/s.

Minor Comments

  1. The authors use a full stop and then add references, it was wrong. Please correct it in the whole MS.
  2. Interval cancers: So what would be the reason for not being detected?
  3. Cited figure 1 within the main text of the manuscript.
  4. Write a specific legend for figure 2.
  5. In table one style of reference is not correct.
  6. The full form of DBT should be mentioned only once in MS and use abbreviated forms everywhere. 
  7. Table 1 shows no recent data of the study point. Please include a few new data if available.
  8. In my opinion, the study point of data in tale 1 should be in ascending or descending order of the study point. 
  9. Write the citation [43][46] in the same box.
  10. Add few references that are more relevant. 
